## PERSPECTIVE

### From dropping to falling: Nocturnal blood pressure decline and fall risk in older adults

**Audrey P. Stegman**
**and Christopher M. Depner** 🄳

*Department of Health and Kinesiology, University of Utah, Salt Lake City, UT, USA*

Email: christopher.depner@utah.edu

Handling Editors: Laura Bennet & Josiane Broussard

The peer review history is available in the Supporting Information section of this article (https://doi.org/10.1113/JP289779#support-information-section).

About one in three older adults experience falls, especially at night, representing a significant contributor to morbidity and mortality (McMahon et al., 2012). One mechanism potentially contributing to risk of such falls is an impaired blood pressure (BP) response to standing, where decreased BP after standing up leads to a loss of balance. Intriguingly, data from controlled tilt-table tests show the circadian system regulates BP responses to postural changes (Hu et al., 2011). This raises the hypothesis that the circadian clock, which controls 24 h rhythms in physiology, may influence BP responses to standing and, in turn, affect night-time fall risk in older adults. However, BP responses to standing after awakening from sleep have not been systematically studied, especially during the night. This knowledge contributes to the lack of effective evidence-based interventions to mitigate night-time fall risk among older adults.

For example, interventions will probably differ if fall risk is modulated by the circadian clock *vs.* environmentally or behaviourally driven rhythms that typically occur at night, such as fasting, being sedentary and medication use, or an interaction among these factors (Klerman et al., 2022). Disentangling the potential influence of the circadian clock *vs.* environmental/behavioural rhythms requires carefully designed circadian protocols such as the constant routine or forced-desynchrony. In general, these protocols eliminate or evenly distribute factors that can mask or influence the circadian clock, meaning that any detected physiological rhythms are strongly influenced by the circadian clock (Klerman et al., 2022).

Thus, to help define the contribution of the circadian clock to BP responses, and potentially fall risk in older adults, Thosar & Shea (2025) conducted a secondary analysis of a forced-desynchrony protocol where they quantified BP responses to standing across all circadian phases. The protocol used 10 ultra-short sleep-wake cycles of 5 h 20 min (2 h 40 min each of scheduled wakefulness and sleep per cycle), lasting 5 days. Participants consisted of two groups, midlife ($n = 19$; aged 40–59 years) and older age ($n = 6$; aged 60–70 years). Analyses quantified heart rate (HR), systolic BP and diastolic BP at 1 and 3 min in response to standing slowly from a supine position, 24 min into each waking episode. Key results showed the 1 min systolic BP response to standing was stable across all circadian phases in the midlife group, whereas the older group showed significantly depressed systolic BP at 1 min after standing, indicating an impaired BP response. Visually, Thosar & Shea (2025) identified the circadian biological night as the time of day with greatest impairment in the older group. Individual data plots show some variability in this impairment across participants in the midlife and older groups, with more consistent impairments in the older group. These findings uncover an impaired BP response to standing in older adults, especially at night. Because Thosar & Shea (2025) leveraged a forced-desynchrony protocol, a dysregulated circadian rhythm of BP responsivity probably plays a major role in this impairment, and may therefore represent a physiological mechanism contributing to risk of night-time falls among older adults. However, for participant safety, the standing protocol was conducted to ensure participants could not fall, precluding analyses of actual fall risk and highlighting a need for future research to more precisely quantify actual falls.

These findings are novel and will help advance the field by setting the stage for larger follow-up studies. Because night-time falls predict increased morbidity and mortality, this is an important new focus for circadian research with significant potential to improve health and quality of life. Thosar & Shea (2025) note that they did not study autonomic mechanisms and they speculate that impaired peripheral vasoconstriction may be a key underlying mechanism driving their findings. Although not mentioned by Thosar & Shea (2025), older adults may experience decreased circadian amplitude (Duffy et al., 2015). It will thus be important to determine whether impaired BP responses to standing are linked to a potential decreased amplitude of the circadian clock in older adults. Regardless, further research is needed to define the primary mechanism underlying the dysregulated BP response in the older group, which will help inform more precise targets and designs of potential intervention or countermeasure strategies.

Despite the rigor of the forced-desynchrony protocol, there are additional aspects to address in future research. The older group had a relatively small sample size ($n = 6$), with only one female participant. Even for rigorously controlled laboratory studies, this is a small sample, and studies powered to identify potential sex differences are needed. The overall study sample was relatively healthy adults, without prescription medication use or sleep disorders including obstructive sleep apnoea. Because many older adults use medications and obstructive sleep apnoea is common, it will be critical to understand how these factors may interact with, and potentially exacerbate, the dysregulated BP response in the older group. If these or other factors do exacerbate the potential risk of falls, it will be important to incorporate such knowledge in the design of new interventions. Despite limited generalizability, it is important to note that this type of basic human physiological research is a critical early step to identify mechanisms we can target in potential interventions or countermeasures. The findings reported by Thosar & Shea (2025) represent an important first step in applying circadian protocols to higher-risk populations and lay the foundation for future research targeting circadian regulation of BP responses to standing as a strategy to reduce fall risk.

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

## Additional information

### Competing interests

No competing interests declared.

### Author contributions

A.P.S. and C.M.D. both contributed significantly to the work. C.M.D. drafted the initial manuscript, and both authors revised it critically for important intellectual content. Both authors approved the final version of the manuscript submitted for publication and agree to be accountable for all aspects of the work in ensuring that questions related to the accuracy or integrity of any part of the work are appropriately investigated and resolved.

### Funding

CMD reports funding from the NIH and the Ben B. and Iris M. Margolis Foundation that is unrelated to this work. APS reports funding from the University of Utah Diabetes and Metabolism Research Center and Labfront that is unrelated to this work.

## Keywords

aging, balance, blood pressure, circadian rhythm, sleep

## Supporting information

Additional supporting information can be found online in the Supporting Information section at the end of the HTML view of the article. Supporting information files available:

**Peer Review History**

