## [Peer Review History · The Journal of Physiology]

From Dropping to Falling: Nocturnal Blood Pressure Decline and Fall Risk in Older Adults

Audrey P Stegman and Christopher M Depner
DOI: 10.1113/JP289779

Corresponding author(s): Christopher Depner (christopher.depner@utah.edu)

The following individual(s) involved in review of this submission have agreed to reveal their identity: Saurabh S Thosar (Referee #1)

Review Timeline:

Submission Date:	06-Aug-2025
Editorial Decision:	21-Aug-2025
Revision Received:	21-Aug-2025
Accepted:	10-Sep-2025

Senior Editor: Laura Bennet

Reviewing Editor: Josiane Broussard

Transaction Report:

Dear Dr Depner,

Re: JP-P-2025-289779 "**From Dropping to Falling: Nocturnal Blood Pressure Decline and Fall Risk in Older Adults**"
by Audrey P Stegman and Christopher M Depner

Thank you for submitting your manuscript to The Journal of Physiology. It has been assessed by a Reviewing Editor and by 1 expert referee and we are pleased to tell you that it is acceptable for publication following satisfactory revision.

The review comments are copied at the end of this email.

Please address all the points raised and incorporate all requested revisions or explain in your Response to Referees why a change has not been made. We hope you will find the comments helpful and that you will be able to return your revised manuscript within 2 weeks. If you require longer than this, please contact journal staff: jp@physoc.org.

REVISION CHECKLIST:

We look forward to receiving your revised submission.

Yours sincerely,

Laura Bennet
Senior Editor
The Journal of Physiology

EDITOR COMMENTS

Reviewing Editor:

Thank you for this clear and well-written perspective. Minor suggestions are provided to improve accuracy and clarity.

REFEREE COMMENTS

Referee #1:

This is a very nicely written perspective. We have some suggestions:

1) Instead of referencing Thosar et al, 2018, JCI, please use the original paper for the tilt-table discussion: Hu, Kun, et al. "Endogenous circadian rhythm in vasovagal response to head-up tilt." *Circulation* 123.9 (2011): 961-970.

2) There is a typo on the length of this FD. The correct FD length is 5 h 20 min with 2 h 40 min cycles of sleep opportunity and standardized wake episodes.

3) It is unlikely that the results are due to a dysregulated circadian timing. This protocol is conducted in dim light, which allows the circadian pacemaker to free run, without dysregulating the circadian system as can happen in overnight protocols conducted in bright light. We can thus isolate the endogenous circadian clock's contributions. Nonetheless, this is a commentary, and it is their prerogative to hypothesize this as a possible mechanism.

END OF COMMENTS

Dear Dr. Bennet and Broussard

We thank the reviewers for their insightful and constructive comments on our manuscript (JP-P-2025-289779) and for the opportunity to revise this manuscript. We have addressed the comments below and revised the manuscript accordingly with highlights to denote all edits. We uploaded clean and tracked changes versions of the revised manuscript. We believe the manuscript has improved based on the helpful feedback, and we hope that it is now acceptable for publication.

Sincerely,

Christopher Depner, Ph.D.

EDITOR COMMENTS

Reviewing Editor:

Thank you for this clear and well-written perspective. Minor suggestions are provided to improve accuracy and clarity.

RESPONSE: Thanks so much, please see below point-by-point responses.

REFEREE COMMENTS

Referee #1:

This is a very nicely written perspective. We have some suggestions:

RESPONSE: Thank you for the helpful comments.

1) Instead of referencing Thosar et al, 2018, JCI, please use the original paper for the tilt-table discussion: Hu, Kun, et al. "Endogenous circadian rhythm in vasovagal response to head-up tilt." *Circulation* 123.9 (2011): 961-970.

RESPONSE: We agree and replaced the Thosar citation with the Hu, Kun citation as suggested.

2) There is a typo on the length of this FD. The correct FD length is 5 h 20 min with 2 h 40 min cycles of sleep opportunity and standardized wake episodes.

RESPONSE: Thank you so much for catching this typo. We made the noted correction.

3) It is unlikely that the results are due to a dysregulated circadian timing. This protocol is conducted in dim light, which allows the circadian pacemaker to free run, without dysregulating the circadian system as can happen in overnight protocols conducted in bright light. We can thus isolate the endogenous circadian clock's contributions. Nonetheless, this is a commentary, and it is their prerogative to hypothesize this as a possible mechanism.

RESPONSE: We agree the results are not likely due to a circadian misalignment, based on the protocol. We had intended to suggest the results may be due to the circadian system having a dysregulated or dysfunctional (inadequate) response or contribution to standing during the biological night in older adults. We revised this portion of paragraph 3 in our manuscript to help improve clarity as follows.

"Because the authors leveraged a forced-desynchrony protocol, a dysregulated circadian rhythm of BP responsiveness likely plays a major role in this impairment, and may therefore represent a physiological mechanism contributing to risk of nighttime falls among older adults."

Dear Professor Depner,

Re: JP-P-2025-289779R1 "**From Dropping to Falling: Nocturnal Blood Pressure Decline and Fall Risk in Older Adults**" by Audrey P Stegman and Christopher M Depner

We are pleased to tell you that your paper has been accepted for publication in The Journal of Physiology.

Yours sincerely,

Laura Bennet
Senior Editor
The Journal of Physiology

If you would like to receive our 'Research Roundup', a monthly newsletter highlighting the cutting-edge research published in The Physiological Society's family of journals (The Journal of Physiology, Experimental Physiology, Physiological Reports, The Journal of Nutritional Physiology, and The Journal of Precision Medicine: Health and Disease), please click this link, fill in your name and email address and select 'Research Roundup':

<https://www.physoc.org/journals-and-media/membernews>

- You can help your research get the attention it deserves! Check out Wiley's free Promotion Guide for best-practice recommendations for promoting your work at: www.wileyauthors.com/eeo/guide. You can learn more about Wiley Editing Services which offers professional video, design, and writing services to create shareable video abstracts, infographics, conference posters, lay summaries, and research news stories for your research at: www.wileyauthors.com/eeo/promotion.

The Corresponding Author will receive an email from Wiley with details on how to register or log-in to Wiley Authors Services where you will be able to place an order

EDITOR COMMENTS

Reviewing Editor:

Thank you for this responsive resubmission. There are no further suggestions for revisions.

REFEREE COMMENTS

Referee #1:

Thanks for addressing our suggestions.